# The Ubiquitiformal Characterization of the Mesostructures of Polymer-Bonded Explosives

**DOI:** 10.3390/ma12223763

**Published:** 2019-11-15

**Authors:** Yibo Ju, ZhuoCheng Ou, Zhuoping Duan, Fenglei Huang

**Affiliations:** School of Mechatronical Engineering, Beijing Institute of Technology, Beijing 100081, China; sdwhsssb1991@163.com (Y.J.); duanzp@bit.edu.cn (Z.D.); huangfl@bit.edu.cn (F.H.)

**Keywords:** nesting ubiquitiform model, complexity, PBX, particle size distribution, thermal conductivity

## Abstract

A nesting ubiquitiform (NU) approach was developed to characterize the mesostructural features of polymer-bonded explosives (PBXs), and then used to predicate some equivalent physical properties of PBXs, which can also be expected to be extended to other composites with complicated internal mesostructures. To verify the availability, two NU models for two kinds of PBX with different compositions are presented, which are PBX 9501 and LX-17, based on which, the equivalent thermal conductivities were calculated. Particularly, it is so encouraging that an analytical expression of the equivalent thermal conductivity was obtained only under a simply assumption of homogeneity. Moreover, it was found that the numerical results calculated by both the recursive algorithm and the analytical expression were in good agreement with the experimental data. In addition, it is also shown that such a physical property as the equivalent thermal conductivity is indeed independent of the meso-configuration of the location distribution of the explosive particles and the voids inside the PBX, which seems consistent with the common expectations and lays the foundations for the application of ubiquitiform to investigating some equivalent properties of composites.

## 1. Introduction

Polymer-bonded explosives (PBXs), in which the explosive powder is bound together with the matrix, have been widely used in both the civil and the military engineering applications, due to advantages such as easy shaping and safe machining. To better understand the macroscopic chemical and mechanical behaviors of a PBX, such as its decomposition, combustion, detonation, constitutive behaviors, and so on, it is especially important to characterize its mesostructural features. However, it is very difficult to describe, reasonably, the mesostructure of any PBX due to their significant heterogeneity and structural complexity, which has received much academic attention over the past decades [1,2,3,4] and is still a challenge now.

On the other hand, since the pioneering work of Mandelbrot [5,6], which introduced an effective nonlinear mathematical tool, fractal has been extensively used in investigating the mesostructure features of various kinds of the quasi-brittle materials, such as concrete [7], rock [8], and soil [9,10]. Among them, Carpinteri et al. [7] derived the fractal patterns in the tensile failure of concrete specimens from the grain size distributions of the aggregates inside the material, and proposed a fractal cohesive crack model, by which the size-independence of the fracture energy was proven; Katz and Thompson [8] showed that the pore spaces formed by several sandstones are fractal, and they predicted the correct porosity of the rock by using the principle of fractal statistics. Tyler and Wheatcraft [9,10] presented an analytical soil water retention model based on the fractal characteristics of soil, and the numerical results estimated by the model are in good agreement with the experimental data. Moreover, by using an appropriate map from a pre-fractal domain to a continuum region, Davey and Prosser [11] obtained an analytical solution for a steady-state heat transfer problem on a fractal domain. Recently, the concept of fractal has also been used for PBX. For example, Cheng et al. [12] carried out a fractal analysis on the morphology of a Triaminotrinitrobenzene (TATB)-based atomic force microscopy (AFM) explosive. It was shown that the binder chains were gradually activated with the increasing temperature, which resulted in a spread of binder and decrease in the fractal dimension of the internal surface of the explosive. Mang and Hjelm [13] used a small-angle neutron scattering technique to study the fractal network formed by the inter-granular voids in a pressed TATB explosive. It was found that the volume fractal dimension can be used to characterize the voids arrangement and that the fractal analysis on the surface area can be used to quantify the average TATB grain size. Moreover, it is particularly noteworthy that both the studies showed that the corresponding fractal dimension can be determined uniquely by the particle gradation of the explosive samples, which implies that the macroscale physical properties of the PBXs should be related to their mesoscale fractal structure, and hence, can be described by using the fractal dimension.

However, conceptually speaking, it seems that some inherent difficulties appear with the development of fractal applications, especially when involving the fractal measure of the “fractal objects” under consideration, due to the singularity of the integral dimensional measure of a fractal. Recently, a new concept of ubiquitiform has been proposed by Ou et al. [14], according to which, any physical or geometrical object in nature is of ubiquitiform, rather than a fractal. Moreover, the fractal approximation of such a ubiquitiform is unreasonable, because of the divergence of the integral dimensional measure of the fractal. Certainly, the concept of a ubiquitiform is also much different from that of a “prefractal,” in that, although both of them are generated through a finite iteration procedure, conceptually, the latter is still a fractal with a definite fractal dimension, while the former, by definition, is of integral dimension with a definite fractional complexity. So far, the concept of ubiquitiform has been used successfully to describe some physical properties of composite materials. For example, Li et al. [15] proposed a ubiquitiformal, one-dimensional, steady-state conduction model for a cellular material rod, by which, the explicit analytical expressions for both the temperature distribution and the equivalent thermal conductivity are obtained. Ou et al. [16] presented both the conception and the explicit expression of the so-called ubiquitiformal fracture energy for quasi-brittle materials, and found that there is no size effect for the ubiquitiformal fracture energy, which implies that the ubiquitiformal fracture energy will be one of the reasonable fracture parameters. In addition, Li et al. [17] established a statistical ubiquitiformal crack extension model for granular composite materials to describe the ubiquitiform properties of the crack extension paths or of the fracture surfaces.

From the above description, it appears that the ubiquitiform rather than the fractal should be used to describe the mesoscale structure of composite materials as well as the internal mesoscale morphology of PBXs. At the briefest glance, the ubiquitiformal characterization of the mesoscale structure of a PBX will be similar to that of the quasi-brittle materials described above [15,16,17]. However, it should be noticed that there exists a crucial difference in the mesoscale structure between the general quasi-brittle materials and the PBXs. That is, in general, the volume distribution of the aggregates with the aggregate size in a quasi-brittle material is simply of “single peak”, which can be characterized well by a single ubiquitiform (SU) model with one constant ubiquitiform complexity over the total mesoscale configuration. While, in a PBX (see the figure 8.1 in the reference [18] for example), typically, the explosive particles are divided into two groups according to the particle sizes; namely, the coarse particles (mean diameter of ~550 μm) and the fine particles (mean diameter of ~25 μm). Just as presented by Baer [18], a PBX is that a bimodal mixture of High melting explosive (HMX) crystals forms a skeletal matrix (the coarse particles) with a closely-packed configuration, with the fine crystallites of HMX (the fine particles) filled in the interstices between the large grains. Such a nesting mesoscale structure usually forms a “double peak” volume distribution of the explosive particles [18], which is difficult to be described by using an SU model, as is done for the quasi-brittle materials [15,16,17]. In the opinion of the present authors, such a nesting configuration may be described better by using a nesting ubiquitiform (NU) structure, one ubiquitiform nested in another, and the domains occupied by the two particle groups which are characterized, respectively, by two different SU models or sub-ubiquitiforms with different ubiquitiform complexities.

The purpose of this study was to develop an NU model to characterize the mesostructure of PBX, based on which, two NU models for two different kinds of PBX, PBX 9501 and LX-17, respectively, were proposed. The compositions of those two PBXs are introduced in Section 4. To verify the availability, the numerical results predicted by using the NU characterization for both the particle size distribution (PSD) and the equivalent thermal conductivity of the two explosives were compared with the previous experimental data, showing good agreement. This paper is divided into the following five sections. After this brief introduction, the nesting ubiquitiformal characterization on the mesostructure of PBX is described in detail in Section 2, and then the NU model is used to obtain the equivalent thermal conductivity of a PBX in Section 3. In Section 4, the numerical results are presented and compared with the previous experimental data, together with some discussions. Finally, in Section 5, some conclusions are drawn out.

## 2. Nesting Ubiquitiform Model for the Mesostructure of PBX

As is well known, any self-similarity of the internal mesostructure of material is available only in a finite scale range of [*δ*_min_, *δ*_max_], where *δ*_min_ and *δ*_max_ are the lower and upper bounds to the scale invariance of the “yardstick” used for the measurement, respectively. Such self-similarity can usually be described by using the so-called ubiquitiform. In general, a ubiquitiform with a single complexity *D* can be used to characterize the mesoscale configuration of a composite material with an ideal “single peak” volume distribution of the inclusions. However, as described in the previous section, PBX has an obvious “double peak” volume distribution of the explosive particles, together with a volume distribution of the voids, which implies that the distributions of explosive particles can be classified into two classes; i.e., the coarse and the fine one. Hence, a nesting ubiquitiform (NU) model was proposed in this study to characterize the mixed mesostructure composed of the coarse and the fine explosive particles and the voids. A NU model is defined as an *N*-layers (*N* > 1) nesting sub-structures, and each of the sub-structures is assumed to be a ubiquitiform with a single complexity, which will be called “sub-ubiquitiform” hereinafter. Thus, for the *k*-th (*k* = 1, 2, …, *N*) sub-ubiquitiform in a NU model, the *k*-th self-similarity is available at the scale interval [$\delta_{\min}^{\left(k\right)}$, $\delta_{\max}^{\left(k\right)}$], with the number of iterations *n_k_* and the ubiquitiform complexity *D_k_*. Without loss of generality, it is assumed that
(1)$$\delta_{\max}^{\left(k+1\right)}\leq \delta_{\min}^{\left(k\right)}\;\left(k=1,2,\cdots ,N-1\right)$$

To characterize the mesostructure of a PBX by using a NU model, each of the sub-ubiquitiforms will be described by using ubiquitiformal generalized Menger sponges with certain geometric properties. That is, for the *k*-th sub-ubiquitiform, a cube with the side length of $\delta_{\max}^{\left(k\right)}$ is taken as its initial element, and then, in the *i*-th (*i* ≤ *n_k_*) iteration step, the reserved cube is subdivided into *p_k_*^3^ sub-cubes with the same side length $\delta_{i}^{\left(k\right)}$, among which, *q_k_* (*q_k_*/*p_k_*^3^ < 1) sub-cubes are removed. The removed sub-cubes represent the inclusions inside the PBX under consideration. As an example, the configuration of a two-dimensional, two-layer NU model is depicted in Figure 1, in which the black areas are the removed sub-squares that represent the inclusions of the PBX, and the reserved grey squares will be subdivided in the next iteration. Moreover, Figure 1a shows the configuration of the first iteration of the first sub-ubiquitiform, and Figure 1b, the overall configuration of the two-layer NU model, in which, each of the reserved grey squares with black dots represents the second sub-ubiquitiform, as shown in the local enlarged schematic diagram on the right.

According to Ou et al. [14], the complexity *D_k_* of the *k*-th sub-ubiquitiform is
(2)$$D_{k}=\frac{\log\left(p_{k}^{3}-q_{k}\right)}{\log p_{k}}\;\left(k=1,2,\cdots ,N\right).$$

In this study, the scale factor *p_k_* of the *k*-th sub-ubiquitiform model is taken as the ratio of the upper bound of the scale invariance $\delta_{\max}^{\left(k\right)}$ to the maximum inclusion size $d_{\max}^{\left(k\right)}$; i.e.,
(3)$$p_{k}=\frac{\delta_{\max}^{\left(k\right)}}{d_{\max}^{\left(k\right)}}\;\left(k=1,2,\cdots ,N\right).$$

Subsequently, the side length $\delta_{i}^{\left(k\right)}$ of the inclusions generated in each iteration can be calculated by
(4)$$\delta_{i}^{\left(k\right)}=\frac{\delta_{\max}^{\left(k\right)}}{p_{k}^{i}}\;\left(i=1,2,\cdots ,n_{k};\;k=1,2,\cdots ,N\right).$$

Hereinafter, for convenience, note that $\delta_{0}^{\left(k\right)}=\delta_{\max}^{\left(k\right)}$, $\delta_{1}^{\left(k\right)}=d_{\max}^{\left(k\right)}$. Thus, the number of iterations *n_k_* can be written as
(5)$$n_{k}=\frac{\log\left(\delta_{\max}^{\left(k\right)}⁄\delta_{\min}^{\left(k\right)}\right)}{\log p_{k}}\;\left(k=1,2,\cdots ,N\right).$$

Eventually, the last parameter of the *k*-th sub-ubiquitiform model needed to be determined is *q_k_*, which will be related to the PSD of the *k*-th sub-ubiquitiform. In most cases, the inclusion distribution in a PBX is not strictly self-similar. As a result, it is almost impossible to find an appropriate NU model that has exactly the same PSD as that of the PBX. Hence, what one can expect is to make the PSD of the NU model close to that of the PBX as far as is possible. According to the extreme value theorem, the “minimum error” is taken for determining *q_k_*; that is,
(6)$$\frac{\partial Err_{q}^{\left(k\right)}}{\partial q_{k}}=0,\;\frac{\partial^{2}Err_{q}^{\left(k\right)}}{\partial q_{k}^{2}}>0\;\left(1\leq k\leq N\right),$$
where ${Err}_{q}^{\left(k\right)}$ is the error between the PSD calculated by using the *k*-th sub-ubiquitiform and that of the PBX based on the theory of Kullback–Leibler divergence function (also called relative entropy) [19], as
(7)$$Err_{q}^{\left(k\right)}=\sum_{i=1}^{n_{k}}\left|V_{\mathrm{inc}}^{\left(e\right)}\left(\delta_{i}^{\left(k\right)}\right)\log\left(\frac{V_{\mathrm{inc}}^{\left(e\right)}\left(\delta_{i}^{\left(k\right)}\right)}{V_{\mathrm{inc}}^{\left(u\right)}\left(\delta_{i}^{\left(k\right)},q_{k}\right)}\right)\right|\;\left(1\leq i\leq n_{k}\right),$$
where $V_{\mathrm{inc}}^{\left(e\right)}$($\delta_{i}^{\left(k\right)}$) is the experimentally measured total volume of the inclusions with size $\delta_{i}^{\left(k\right)}$ inside the PBX, and $V_{\mathrm{inc}}^{\left(u\right)}$($\delta_{i}^{\left(k\right)}$, *q_k_*) is the corresponding total volume of the inclusions with size $\delta_{i}^{\left(k\right)}$ of the NU model. According to the iteration rule, in *k*-th sub-ubiquitiform, $V_{\mathrm{inc}}^{\left(u\right)}$($\delta_{i}^{\left(k\right)}$, *q_k_*) can be defined as
(8)$$V_{\mathrm{inc}}^{\left(u\right)}\left(\delta_{i}^{\left(k\right)},q_{k}\right)=V_{\mathrm{rs}}^{\left(u\right)}\left(\delta_{i-1}^{\left(k\right)},q_{k}\right)-V_{\mathrm{rs}}^{\left(u\right)}\left(\delta_{i}^{\left(k\right)},q_{k}\right)\;\left(i=1,2,\cdots ,n_{k};k=1,2,\cdots ,N\right),$$
where $V_{\mathrm{rs}}^{\left(u\right)}$($\delta_{i-1}^{\left(k\right)}$, *q_k_*) and $V_{\mathrm{rs}}^{\left(u\right)}$($\delta_{i}^{\left(k\right)}$, *q_k_*) are the total volumes of the reserved sub-cubes after the (*i*−1)-th and *i*-th iterations, respectively. In fact, the volume of the initial element of the *k*-th sub-ubiquitiform is equal to the volumes of the reserved sub-cubes of the (*k* − 1) sub-ubiquitiform, which can be denoted as $V_{\mathrm{rs}}^{\left(u\right)}$($\delta_{\min}^{\left(k-1\right)}$, *q_k_*_−1_). Thus, $V_{\mathrm{rs}}^{\left(u\right)}$($\delta_{i}^{\left(k\right)}$, *q_k_*) can be presented as
(9)$$V_{\mathrm{rs}}^{\left(u\right)}\left(\delta_{i}^{\left(k\right)},q_{k}\right)=V_{\mathrm{rs}}^{\left(u\right)}\left(\delta_{\min}^{\left(k-1\right)},q_{k-1}\right){\left(\frac{p_{k}^{3}-q_{k}}{p_{k}^{3}}\right)}^{i}\;\left(i=1,2,\cdots ,n_{k};\;k=1,2,\cdots ,N\right).$$
From this recursive relationship, $V_{\mathrm{rs}}^{\left(u\right)}$($\delta_{i}^{\left(k\right)}$, *q_k_*) can be further rewritten as
(10)$$V_{\mathrm{rs}}^{\left(u\right)}\left(\delta_{i}^{\left(k\right)},q_{k}\right)=V_{0}{\left(\frac{p_{k}^{3}-q_{k}}{p_{k}^{3}}\right)}^{i}\prod_{j=1}^{k-1}{\left(\frac{p_{j}^{3}-q_{j}}{p_{j}^{3}}\right)}^{n_{j}}\left(i=1,2,\cdots ,n_{k};\;k=1,2,\cdots ,N\right),$$
where *V*_0_ is the total volume of the initial element of the NU model. Substituting Equation (10) into Equation (8), the PSD function of the *N*-layers NU model can be obtained as
(11)$$V_{\mathrm{inc}}^{\left(u\right)}\left(\delta_{i}^{\left(k\right)},q_{k}\right)=V_{0}\left[{\left(\frac{p_{k}^{3}-q_{k}}{p_{k}^{3}}\right)}^{i-1}-{\left(\frac{p_{k}^{3}-q_{k}}{p_{k}^{3}}\right)}^{i}\right]\;\prod_{j=1}^{k-1}{\left(\frac{p_{j}^{3}-q_{j}}{p_{j}^{3}}\right)}^{n_{j}}\left(i=1,2,\cdots ,n_{k};\;k=1,2,\cdots ,N\right)$$

So far, all parameters needed for establishing each of the sub-ubiquitiforms and the NU model have been determined. The next step is to get the configuration of the NU model; namely, the concrete spatial locations of the inclusions distribution in the matrix. As above described, in a *N*-layers NU model, the positions of the sub-cubes removed in each iteration are used to indicate the locations of the explosive particles inside the PBX. Therefore, to establish a NU model for better reflecting the chemical and physical properties of a PBX, the concrete locations of the removed sub-cubes in each iteration step must be determined; only by that could a reasonable spatial distribution of the explosive particles be obtained.

Obviously, in general, the spatial distribution of the explosive particles inside a PBX is random. Therefore, to better characterize such a random feature, in this study, the sub-cubes to be removed were chosen stochastically among the all sub-cubes. In other words, in the first iteration of the *k*-th sub-ubiquitiform, each initial cube was divided into $p_{k}^{3}$ sub-cubes, among which the concrete positions of the *q_k_* sub-cubes removed were distributed stochastically. As an example, a random configuration of a *k*-th sub-ubiquitiform is depicted as in Figure 2.

Moreover, to ensure the strict self-similarity of the *k*-th sub-ubiquitiform, the removing style or the positional distribution of the *q_k_* sub-cubes in the first iteration remained unchanged when applying the subsequent iteration process of the *k*-th sub-ubiquitiform model. Certainly, because of the randomness, there exists a certain degree of uncertainty on the concrete location distribution of the explosive particles in a NU model, and hence, it seems that the rationality for the practical applications of the NU model must be taken into account, especially when it comes to predicting the equivalent properties of the model (such as the equivalent Young’s modulus, the equivalent thermal conductivity, and so on). That is to say, can the NU model with the same model parameters but different configurations lead to equivalent results? Are the equivalent properties independent of the specific ubiquitiform configuration? Indeed, it is still a challenge to answer this question theoretically. However, in practice, it seems not so serious a problem, because of the fact that, under the same gradation of inclusions and the same mechanical properties of both the matrix and the inclusions, the composite materials produced in different batches show basically the same equivalent mechanical properties, which is also the theoretical foundation of analyzing the whole properties of a multi-phase material by any random approach. In other words, in the opinion of the present authors, a NU model is an especially typical structure among all of the random configurations of the location distributions of the inclusions, which can be used to represent the mesostructure of the granular composite material under consideration in predicting the material’s equivalent chemical and physical properties.

An *N*-layers NU model for a PBX can be established by giving all the parameters *p_k_*, $\delta_{\max}^{\left(k\right)}$, *n_k_*, and *q_k_* (*k* = 1, 2, …, *N*). For the sake of clarity, the determination steps of these model parameters can be summarized as follows:
Based on the *N* different types of the explosive particles (the corase and the fine ones, for instance), the PBX under consideration is classified first into the *N* sub-material groups nested one by one. Each of the sub-material groups are described by using a sub-ubiquitiform, and all the sub-ubiquitiforms that are nested, one by one form an *N*-layers NU model. For the *k*-th sub-ubiquitiform, the upper bound to scale invariance of the “yardstick” used for the measurement $\delta_{\max}^{\left(k\right)}$ can usually be assumed to be the size of the representative volume element (RVE) of the *k*-th sub-material group, which is usually taken to be 5–10 times the maximum particle size in the *k*-th sub-material group. On the other hand, the lower bound to scale invariance $\delta_{\min}^{\left(k\right)}$ can be taken to equal to the minimum particle size in the *k*-th sub-material group;The scaling factor *p_k_* is determined by Equation (3);The number of iterations *n_k_* is determined from Equation (5);The number of the removed sub-cubes *q_k_* is determined by Equation (6);The complexity *D_k_* of the *k*-th sub-ubiquitiform model can be obtained from Equation (2).

## 3. Equivalent Thermal Conductivity of PBX

To show the practical application of the NU model, the equivalent thermal conductivity of a PBX will be analyzed in this section. The position-space renormalization group technique [20] is used here to calculate the equivalent thermal conductivity $K_{\mathrm{equ}}^{\left(k\right)}$ of the *k*-th sub-ubiquitiform. In the following, a recursive relationship between the equivalent thermal conductivity $K_{\mathrm{rs}}^{\left(i-1,k\right)}$ of each reserved sub-cube of the (*i*−1)-th iteration and that of the *i*-th iteration $K_{\mathrm{rs}}^{\left(i,k\right)}$ is obtained first, and then the total equivalent thermal conductivity of the *k*-th sub-ubiquitiform $K_{equ}^{\left(k\right)}$ ≡ $K_{\mathrm{rs}}^{\left(0,k\right)}$ can be derived directly through a recursive approach.

To obtain the recursive relation between $K_{\mathrm{rs}}^{\left(i-1,k\right)}$ and $K_{rs}^{\left(i,k\right)}$, see Figure 3 as an example. Firstly, a reserved sub-cube of the (*i*−1)-th iteration is iterated into the sub-cubes of the *i*-th iteration with a certain configuration, as shown in the left image of Figure 3a with *p_k_* = 3 and *q_k_* = 9. The reserved sub-cube of the (*i−*1)-th iteration is divided into $p_{k}^{3}$ sub-cubes of the *i*-th iteration, among which *q_k_* sub-cubes that represent the inclusions are removed. Next, the configuration of the *i*-th iteration is further separated along the *x*-axis direction into *p_k_* units denoted by unit-1, unit-2, …, unit-*p_k_*, respectively; see the fore-and-aft three units, as shown by the right-hand image in Figure 3a and the three corresponding intersecting surfaces of these three units shown in Figure 3b. Thus, the remaining sub-cubes of the (*i*−1)-th iteration can be seen as a tandem connection of the *p_k_* units, and, in each of the units, a series of the sub-cubes (including both the reserved and the removed sub-cubes) in the configuration of the *i*-th iteration are parallelly connected. In addition, as shown in Figure 3b, the number of sub-cubes of the *j*-th unit removed is denoted by $n_{j}^{\left(k\right)}$(*j* = 1, 2, …, *p_k_*).

The equivalent thermal conductivity of the *j*-th unit is denoted by $K_{i-1}^{\left(j,k\right)}$, which can be calculated by the theory of thermal circuit, as
(12)$$K_{i-1}^{\left(j,k\right)}=\frac{p_{k}^{2}-n_{j}^{\left(k\right)}}{p_{k}^{2}}K_{\mathrm{rs}}^{\left(i,k\right)}+\frac{n_{j}^{\left(k\right)}}{p_{k}^{2}}K_{\mathrm{inc}}^{\left(k\right)}\left(j=1,2,\cdots ,p_{k};\;i=1,2,\cdots ,n_{k};\;k=1,2,\cdots ,N\right),$$
where $K_{\mathrm{inc}}^{\left(k\right)}$ is the thermal conductivity of the inclusions in the *k*-th sub-material group. Then the equivalent thermal conductivity $K_{\mathrm{rs}}^{\left(i-1,k\right)}$ can be obtained by the tandem connection of the *p_k_* units:(13)$$K_{\mathrm{rs}}^{\left(i-1,k\right)}=\frac{p_{k}}{\sum_{j=1}^{p_{k}}\frac{1}{K_{i-1}^{\left(j,k\right)}}}\left(i=1,2,\cdots ,n_{k};\;k=1,2,\cdots ,N\right).$$

Substituting Equation (12) into Equation (13), the recursive relation between $K_{\mathrm{rs}}^{\left(i-1,k\right)}$ and $K_{\mathrm{rs}}^{\left(i,k\right)}$ can be written as
(14)$$K_{\mathrm{rs}}^{\left(i-1,k\right)}=\frac{1}{\sum_{j=1}^{p_{k}}\frac{p_{k}}{\left[(p_{k}^{2}-n_{j}^{\left(k\right)})K_{\mathrm{rs}}^{\left(i,k\right)}+n_{j}^{\left(k\right)}K_{\mathrm{inc}}^{\left(k\right)}\right]}}\left(i=1,2,\cdots ,n_{k};\;k=1,2,\cdots ,N\right).$$

Moreover, according to the iteration rule, the thermal conductivity of the reserved part of the *k*-th sub-ubiquitiform is equal to that of the (*k* + 1) sub-ubiquitiform; that is,
(15)$$K_{\mathrm{rs}}^{\left(n_{k},k\right)}=K_{\mathrm{rs}}^{\left(0,k+1\right)}\left(k=1,2,\cdots ,N\right).$$

Finally, at the end of the overall iteration process, the reserved part can be considered as only the matrix, and its thermal conductivity is then
(16)$$K_{\mathrm{rs}}^{\left(n_{N},N+1\right)}=K_{m},$$
where *K*_m_ is the thermal conductivity of the matrix of the PBX.

So far, with Equations (14)–(16), the equivalent thermal conductivity $K_{\mathrm{rs}}^{\left(0,1\right)}$ of the NU model can be obtained iteratively.

Moreover, it is very encouraging that, under some adequate approximations, an analytical expression for the average equivalent thermal conductivity of the PBXs can be obtained from the *N*-layers NU model. For example, it is believed from the practical production process of the PBXs that the structure of the NU model should be of homogeneous as far as possible, which implies that on average, $n_{1}^{\left(k\right)}$ = $n_{2}^{\left(k\right)}$ = … = $n_{p}^{\left(k\right)}$ ≡ $n_{a}^{\left(k\right)}$(*k* =1, 2, …, *N*) for each sub-ubiquitiform. Thus, for the *k*-th sub-ubiquitiform and denoting that the average equivalent thermal conductivity of the reserved sub-cubes of the *i*-th iteration as $K_{\mathrm{rs},\mathrm{average}}^{\left(i,k\right)}$, the recursive relation Equation (14) can be simplified as
(17)$$K_{\mathrm{rs},\mathrm{average}}^{\left(i-1,k\right)}=A_{k}K_{\mathrm{rs},\mathrm{average}}^{\left(i,k\right)}+B_{k}\;\left(i=1,2,\cdots ,n_{k};\;k=1,2,\cdots ,N\right),$$
where,
(18)$$A_{k}=1-\frac{n_{a}^{\left(k\right)}}{p_{k}^{2}},\;B_{k}=\frac{n_{a}^{\left(k\right)}}{p_{k}^{2}}K_{inc}^{\left(k\right)}\;\left(k=1,2,\cdots ,N\right).$$

Notice that $n_{1}^{\left(k\right)}$ + $n_{2}^{\left(k\right)}$ + … + $n_{p}^{\left(k\right)}$ = *q_k_* (*k* = 1, 2, …, *N*), and then
(19)$$n_{a}^{\left(k\right)}=\frac{q_{k}}{p_{k}}\;\left(k=1,2,\cdots ,N\right)$$

Substituting Equation (19) into Equation (18) yields
(20)$$A_{k}=1-\frac{q_{k}}{p_{k}^{3}},\;B_{k}=\frac{q_{k}}{p_{k}^{3}}K_{\mathrm{inc}}^{\left(k\right)}\;\left(k=1,2,\cdots ,N\right)$$

Thus, by using the recursive relation Equations (17) and (20), an analytical expression for the average equivalent thermal conductivity of the *k*-th sub-ubiquitiform model, i.e., $K_{\mathrm{rs},\mathrm{average}}^{\left(0,k\right)}$ can be obtained, as
(21)$$K_{\mathrm{rs},\mathrm{average}}^{\left(0,k\right)}=K_{\mathrm{inc}}^{\left(k\right)}+{\left(1-\frac{q_{k}}{p_{k}^{3}}\right)}^{n_{k}}\left(K_{\mathrm{rs},\mathrm{average}}^{\left(n_{k},k\right)}-K_{\mathrm{inc}}^{\left(k\right)}\right)\;\left(k=1,2,\cdots ,N\right).$$

Furthermore, notice that the thermal conductivity of the reserved sub-cubes of the *k*-th sub-ubiquitiform is equal to that of the (*k* + 1) sub-ubiquitiform model; i.e.,
(22)$$K_{\mathrm{rs},\mathrm{average}}^{\left(0,k+1\right)}=K_{\mathrm{rs},\mathrm{average}}^{\left(n_{k},k\right)}\;\left(k=1,2,\cdots ,N-1\right).$$

Hence, from Equations (21) and (22), the overall analytical expression for the average equivalent thermal conductivity of the NU model can be obtained as
(23)$$K_{\mathrm{rs},\mathrm{average}}^{\left(0,1\right)}=K_{\mathrm{rs},\mathrm{average}}^{\left(n_{N},N\right)}\sum_{k=1}^{N}{\left(1-\frac{q_{k}}{p_{k}^{3}}\right)}^{n_{k}}+\sum_{k=1}^{N}\left(\sum_{j=1}^{k-1}{\left(1-\frac{q_{j}}{p_{j}^{3}}\right)}^{n_{j}}-\sum_{j=1}^{k}{\left(1-\frac{q_{j}}{p_{j}^{3}}\right)}^{n_{j}}\right)K_{\mathrm{inc}}^{\left(k\right)},$$
in which
(24)$$K_{\mathrm{rs},\mathrm{average}}^{\left(n_{N},N\right)}=K_{m}.$$

In the following section, the numerical results calculated by both the analytical expression Equation (23) and the direct recursive approach from Equation (14) will be presented and compared with the previous experimental data.

## 4. Numerical Results and Discussion

To calculate the numerical results of the equivalent thermal conductivity of a PBX, its PSD function must be determined beforehand. According to Skidmore et al. [21], the PSD function can be written in the following form
(25)$$v_{\mathrm{inc}}^{\left(e\right)}\left(x\right)=a_{k}\exp\left[\frac{-{\left(\mathrm{lg}x-c_{k}\right)}^{2}}{2w_{k}^{2}}\right]\;\left(k=1,2,\cdots ,N\right),$$
where *a_k_*, *c_k_*, and *w_k_* are the parameters corresponding to the PSD of the explosive particles in the *k*-th sub-material group. In general, the PSD of a NU model is a discrete function as Equation (10), which cannot be compared with the continuous PSD of the PBX directly. To overcome such a difficulty and calculate the error between the PSD of the NU model and of the PBX from Equation (6), the continuous function Equation (25) is transformed—approximated into a discrete function in the following form
(26)$$V_{\mathrm{inc}}^{\left(e\right)}(\delta_{i}^{\left(k\right)})=\int_{\delta_{i+1}^{\left(k\right)}}^{\delta_{i}^{\left(k\right)}}v_{\mathrm{inc}}^{\left(e\right)}\left(x\right)\;dx\;\left(i=1,2,\cdots ,n_{k}-1;\;k=1,2,\cdots ,N\right),$$
where $\delta_{i}^{\left(k\right)}$ is determined directly from Equation (4).

Furthermore, the upper bound to scale invariance $\delta_{\max}^{\left(k\right)}$, the lower bound to scale invariance $\delta_{\min}^{\left(k\right)}$, and the maximum inclusion size $d_{\max}^{\left(k\right)}$ can be determined from the PSD function in the following way.

Firstly, according to the experimantal data [21], Figure 4 can be drawn as follows

In Figure 4 the *k*-th PSD and (*k* + 1) PSD of the PBXs always interact with each other, which indicates that the PSDs of two adjoining sub-ubiquitiforms will join with each other, and we denote the abscissa of the cross point as $\delta_{c}^{\left(k\right)}$. Obviously, the cross point can be seen, approximately, as the end of the *k*-th PSD and the beginning of the (*k* + 1) PSD. Since, in this study the lower bound $\delta_{\min}^{\left(k\right)}$ is taken as the minimum size of the *k*-th inclusions, the *k*-th abscissa of the cross point can be taken as the lower bound to scale invariance of the *k*-th sub-ubiquitiform; namely,
(27)$$\delta_{\min}^{\left(k\right)}=\delta_{c}^{\left(k\right)}\;\left(k=1,2,\cdots ,N\right)$$

Next, Equation (1) implies that the lower bound $\delta_{\min}^{\left(k-1\right)}$ of the (*k*−1)-th SU is larger than the upper bound $\delta_{\max}^{\left(k\right)}$ of the *k*-th SU, which can be rewritten in the following form as
(28)$$\delta_{\min}^{\left(k-1\right)}=m_{k}\delta_{\max}^{\left(k\right)}\;\left(k=1,2,\cdots ,N\right).$$

However, so far, there is no good way to obtain the *m_k_* value of each SU from the experiment. Therefore, in this study, we made an assumption according to the fact that the PSDs of two adjoining sub-ubiquitiforms are joined with each other. The assumption is that each reserved sub-cube of the *k*-th sub-ubiquitiform will be the initial cube of the (*k* + 1) sub-ubiquitiform, which implies that the lower bound to scale invariance of the *k*-th sub-ubiquitiform is equal to the upper bound to scale invariance of the (*k* + 1) sub-ubiquitiform; that is,
(29)$$\delta_{\max}^{\left(k+1\right)}=\delta_{\min}^{\left(k\right)}\;\left(k=1,2,\cdots ,N-1\right).$$

It should be noted that there are two remaining bounds to scale invariance that not included in Equation (29); i.e., $\delta_{\max}^{\left(1\right)}$ and $\delta_{\min}^{\left(N\right)}$. Here, we take the largest upper bound to scale invariance $\delta_{\max}^{\left(1\right)}$ to be the size of the representative volume element of the PBX, and the smallest lower bound to scale invariance $\delta_{\min}^{\left(N\right)}$ to be the size of the minimum inclusions of the PBX.

Finally, the maximum size $d_{\max}^{\left(1\right)}$ of the inclusions can be easily read from the PBX. However, there is a little arbitrarity in selecting the values of $d_{\max}^{\left(k\right)}$ (*k* ≥ 2), which depends on the different discrete methods applied to the continuous PSD function in the form of Equation (25). For convenience, it is assumed here that all the scaling factors *p_k_* (*k* = 1, 2, …, *N*) are the same; namely,
(30)$$p_{1}=p_{2}=p_{3}=\cdots =p_{N},$$
where *p*_1_ can be determined directly from Equation (3).

In the following, the numerical results of the NU model and the equivalent thermal conductivity for both the PBX 9501 and the LX-17 explosives will be presented, respectively.

### 4.1. Numerical Results and Discussion of the PBX 9501

The PBX 9501 is composed mainly of the HMX particles embedded within the matrix Estane/BDNPA-F, which can be seen including three kinds of “inclusions”; i.e., the coarse HMX particles, the fine HMX particles, and voids. The parameters in the PSD function for the PBX 9501 used in this study are given by experiment data [21,22], all of them were listed in Table 1, and the subscripts “1”, “2” and “3” represent “the coarse HMX particles,” “the fine HMX particles,” and “the voids,” respectively.

Moreover, according to Liu [23], the size of the RVE $\delta_{\max}^{\left(1\right)}$ = 1500 μm, and the maximum side length of the HMX particles is $d_{\max}^{\left(1\right)}$ = 500 μm. Then, all the parameters in the NU model for the PBX 9501 can be determined, as listed in Table 2. It can be seen that the error ${Err}_{q}^{\left(k\right)}$ between the PSDs calculated by using the NU model and that of the PBX 9501 are all satisfactorily small.

For the PBX 9501, the experimental values of the thermal conductivities $K_{\mathrm{inc}}^{\left(1\right)}$, $K_{\mathrm{inc}}^{\left(2\right)}$, $K_{\mathrm{inc}}^{\left(3\right)}$, and *K*_m_ for the coarse HMX particles, the fine HMX particles, the voids, and the matrixes, respectively, are all listed in Table 3 [24]. In this study, the experimental thermal conductivity of the PBX 9501 is denoted by *K*; three experimental values from different sources were taken as *K* = 0.453W/mK [24], *K* = 0.454 W/mK [25], and *K* = 0.368 W/mK [26]. Based on these parameters, the equivalent thermal conductivity described by using the NU model can be determined from the recursion approach based on Equation (14). Moreover, to estimate the effect of the randomness of the ubiquitiform configuration or of the spatial distribution of the inclusions on the equivalent thermal conductivity, in this study, one hundred equivalent thermal conductivities under the NU models with different configurations are calculated. The mean value $\bar{K}$ of these one hundred calculated equivalent thermal conductivities and the error *Err**_K_* between *K* and $\bar{K}$ are also listed in Table 3. Furthermore, to verify the consistency of the NU models with different configurations, the standard deviation SDEP (%) of the one hundred calculated equivalent thermal conductivity is estimated by
(31)$$\mathrm{SDEP}=\sqrt{\frac{\sum_{i=1}^{100}{\left(K_{i}-\bar{K}\right)}^{2}}{\left(n-1\right){\bar{K}}^{2}}},$$
where *K_i_* (*i* =1, 2, …, 100) is the equivalent thermal conductivity calculated by using the NU model under the i-th configuration. from one hundred NU models.

In addition, the average equivalent thermal conductivity $K_{\mathrm{rs},\mathrm{average}}^{\left(0,1\right)}$ of the NU model can also be predicated directly by using the analytical expression Equation (23), and the numerical result and the error ${Err}_{K}^{\mathrm{average}}$ between $K_{\mathrm{rs},\mathrm{average}}^{\left(0,1\right)}$ and *K* are also listed in Table 3.

It can be seen that the numerical results for the equivalent thermal conductivity calculated by both the recursion approach and the analytical expression Equation (23) are in agreement with most of the experimental data, which implies the availability of the NU model in characterizing the mesostructure feature of the PBX 9501. Moreover, so small a value of SDEP indicates that the equivalent thermal conductivity is independent of the ubiquitiform configuration of the mesostructure of the PBX 9501, which in fact lays the foundations for the applications of ubiquitiform as well as other random models in investigating some equivalent physical and chemical properties of a composite material.

### 4.2. Numerical Results and Discussion of the LX-17

The LX-17 is composed mainly of the TATB particles embedded within the matrix of Kel-F 800. which can be seen including four kinds of “inclusions”; namely, the coarse TATB particles, the fine TATB particles, and the large and the small voids. The parameters in the PSD function for the LX-17 used in this study are given out by experiment data [27,28], all of them were listed in Table 4, in which, the subscripts “1,” “2,” “3,” and “4” represent the coarse and the fine HMX particles, and the large and the small voids, respectively. Similar to that for the PBX 9501, the model parameters of the NU model for the LX-17 are listed in Table 5, and the thermal conductivities used and the numerical results are listed in Table 6.

It can be seen that the numerical results for the equivalent thermal conductivity of the LX-17 calculated by both the recursion approach and the analytical expression Equation (23) are also in agreement with the experimental data, and hence the same inferences as that for the PBX 9501 can be found.

## 5. Conclusions

A nesting ubiquitiform (NU) model was proposed to characterize the mesostructure feature of PBX and then to predicate the equivalent physical properties of the PBX. As an example, two NU models for the PBX 9501 and the LX-17 samples are presented, based on which, the equivalent thermal conductivites of the two explosives were predicated, and an analytical expression for the equivalent thermal conductivity was obtained under some adequate approximations. The numerical results of the equivalent thermal conductivities for both the PBX 9501 and the LX-17 calculated by a recursion approach and the analytical expression are in good agreement with the previous experimental data. Some more conclusions can be drawn out as follows.
The NU model can be used for characterizing the mesostructure feature of PBX;The equivalent thermal conductivity of a PBX can be predicted well by the NU model;The equivalent thermal conductivity is independent of the ubiquitiformal configuration of the explosive particles and the voids inside the PBX, which lays the foundations of the application of ubiquitiform to investigating other equivalent physical and chemical properties.

## Figures and Tables

**Figure 1 materials-12-03763-f001:**
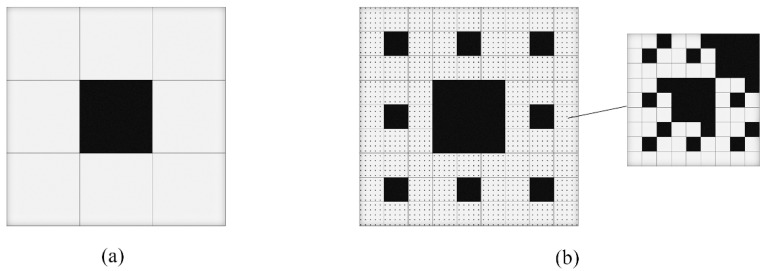
Two-dimensional two-layers nesting ubiquitiform (NU) model with *p*_1_ = 3, *n*_1_ = 2, *q*_1_ = 1 and *p*_2_ = 3, *n*_2_ = 2, *q*_2_ = 2. (**a**) The configuration of the first iteration of the first sub-ubiquitiform (**b**) The overall configuration of the two-layer NU model

**Figure 2 materials-12-03763-f002:**
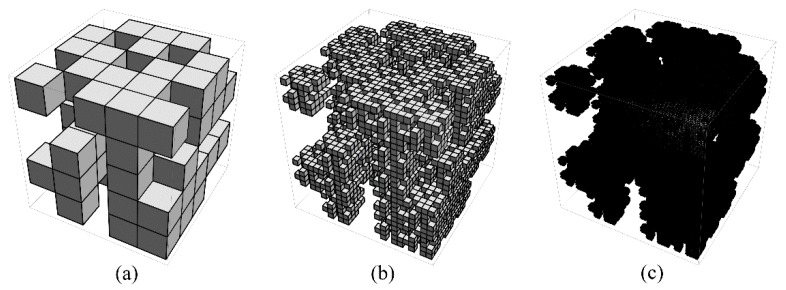
A random configuration of a *k*-th sub-ubiquitiform with *p_k_* = 5, *q_k_* = 64, and *n_k_* = 3. (**a**) The configuration after the first iteration (**b**) The configuration after the second iteration (**c**) The configuration after the third iteration.

**Figure 3 materials-12-03763-f003:**
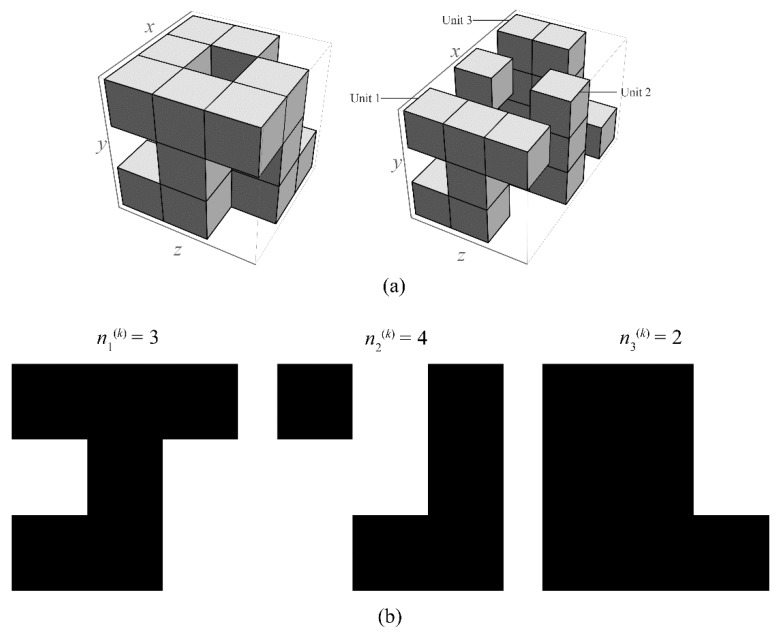
The configuration of the *i*-th iteration of the *k*-th sub-ubiquitiform with *p_k_* = 3 and *q_k_* = 9. (**a**) The structure of the *i*-th iteration. (**b**) The three corresponding intersecting surfaces of the three units.

**Figure 4 materials-12-03763-f004:**
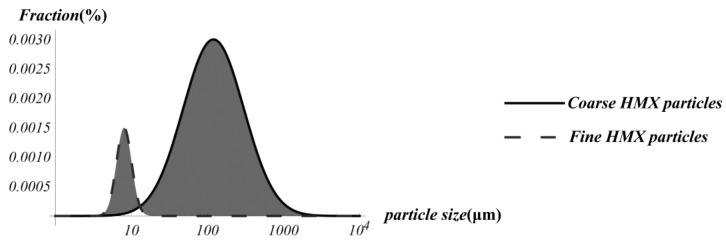
The particle size distribution (PSD) of coarse HMX particles and fine HMX particles of the polymer-bonded explosive (PBX) 9501.

**Table 1 materials-12-03763-t001:** The parameters of the PSD of the PBX 9501.

*a* _1_	*c* _1_	*w* _1_	*a* _2_	*c* _2_	*w* _2_	*a* _3_	*c* _3_	*w* _3_
0.003	2.079	0.4	0.0015	1.176	0.1	0.04	−0.744	0.33

**Table 2 materials-12-03763-t002:** The parameters in the NU model of the PBX 9501.

	*δ*_max_ (μm)	*δ*_min_ (μm)	*p*	*q*	*D*	*n*	${Err}_{q}^{\left(k\right)}$
Coarse HMX	1500	55.5	3	15.4	2.231	3	0.11
Fine HMX	55.5	6.17	3	4.1	2.850	2	6 × 10^−4^
Void	6.17	0.68	3	5.6	2.788	2	0.004

**Table 3 materials-12-03763-t003:** The numerical results for the equivalent thermal conductivity of the PBX 9501.

*K*_m_ (W/m∙K)	$K_{\mathrm{inc}}^{\left(1\right)}$ (W/m∙K)	$K_{\mathrm{inc}}^{\left(2\right)}$ (W/m∙K)	$K_{\mathrm{inc}}^{\left(3\right)}$ (W/m∙K)	$\bar{K}$ (W/m∙K)	$K_{\mathrm{rs},\mathrm{average}}^{\left(0,1\right)}$ (W/m∙K)	*K* (W/m∙K)	${Err}_{K}$	${Err}_{K}^{\mathrm{average}}$	SDEP
0.14	0.51	0.51	0.026	0.489	0.486	0.453	7.95%	7.28%	0.84%
0.14	0.51	0.51	0.026	0.489	0.486	0.454	7.70%	7.04%	0.84%
0.14	0.51	0.51	0.026	0.489	0.486	0.368	32.8%	32%	0.84%

**Table 4 materials-12-03763-t004:** The parameters of the PSD of the LX-17.

*a* _1_	*c* _1_	*w* _1_	*a* _2_	*c* _2_	*w* _2_	*a* _3_	*c* _3_	*w* _3_	*a* _4_	*c* _4_	*w* _4_
0.0204	1.778	0.1	0.0058	1.398	0.2	0.2	−1.699	0.5	0.258	−3	0.3

**Table 5 materials-12-03763-t005:** The model parameters of the NU model of the LX-17.

	*δ*_max_ (μm)	*δ*_min_ (μm)	*p*	*q*	*D*	*n*	${Err}_{q}^{\left(k\right)}$
Coarse TATB	450	50	3	17.3	2.066	2	0.025
Fine TATB	50	5.55	3	3.5	2.872	2	4.4 × 10^−4^
Large voids	5.55	0.0076	3	2.8	2.908	6	8.7 × 10^−3^
Small voids	7.6 × 10^−3^	8.4 × 10^−4^	3	0.3	2.994	2	1.3 × 10^−4^

**Table 6 materials-12-03763-t006:** The numerical result for the equivalent thermal conductivity of the LX-17.

*K*_m_ (W/m∙K)	$K_{\mathrm{inc}}^{\left(1\right)}$ (W/m∙K)	$K_{\mathrm{inc}}^{\left(2\right)}$ (W/m∙K)	$K_{\mathrm{inc}}^{\left(3\right)}$ (W/m∙K)	$K_{\mathrm{inc}}^{\left(4\right)}$ (W/m∙K)	$\bar{K}$ (W/m∙K)	$K_{\mathrm{rs},\mathrm{average}}^{\left(0,1\right)}$ (W/m∙K)	*K* (W/m∙K)	${Err}_{K}$	${Err}_{K}^{\mathrm{average}}$	SDEP
0.052	0.543	0.543	0.026	0.026	0.499	0.494	0.502	0.53%	1.5%	0.37%

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
