# Peer review of "The Ubiquitiformal Characterization of the Mesostructures of Polymer-Bonded Explosives"

_materials, 2019, doi:10.3390/ma12223763_

Round 1

Reviewer 1 Report

The manuscript is well written and explains in detail. I just have a small recommendation to provide literature in depth.

Reviewer 2 Report

This paper deals with characterize of the mesostructure feature of PBX and then used to predicate some equivalent physical properties of PBX by means of a nesting ubiquitiform (NU) approach, which is developed for this purpose in the submitted work.

I consider the chosen methods of the problem solving as logical, including the conclusions drawn from them. The work is really extensive and interesting. However, I lack a comparison of theoretical outputs with published data, which would be confirm a reality of the application of the developed approach in practice. In the case of PBX 9501 I have found this experimental conductivity of 0.368 W/.m.K [1] and of 0.454 W/.m.K [2] while the authors‘ middle value is of 0.489 W/m.K   which is close to the second mentioned value. It seems, that authors‘ approach is capable to give real results. Nevertheless, comparison with experimental values should be included into text and it is valid also for LX-17.

The paper is written clearly and comprehensible. Its conclusions are adequately supported by data (including illustrations and tables) given in its text. A topic of the manuscript is compatible with thematic priorities of the Molecules (Materials)

[1] Jianfeng Lou et al., Numerical Simulation Study on Thermal Response of PBX 9501 to Low Velocity Impact, AIP Conference Proceedings 1793, 030021 (2017); ttps://doi.org/10.1063/1.4971479

[2] J. O. Mares,et al., Thermal and mechanical response of PBX 9501 under contact excitation, Journal of Applied Physics 113, 084904 (2013)

Reviewer 3 Report

This manuscript models the heterogeneous structure of polymer bound explosive materials using a known NU computational model. This reviewer had no major issues with the manuscript and thinks that it may be published in the journal Materials after the authors address some minor issues stated below:

Explicit for of PBX needs to be written in the title.  Also in the abstract explicitly indicate PBX-9501 and LX*-17. what are they? Do not assume readers of Materials know what they are.  PBX or PXB? Correct some errors in the text.  Redo figure 4 with better resolution and dark colors. 
